# Discoloration Investigations of Yellow Lantern Pepper Sauce (*Capsicum chinense* Jacq.) Fermented by *Lactobacillus plantarum*: Effect of Carotenoids and Physiochemical Indices

**DOI:** 10.3390/molecules27207139

**Published:** 2022-10-21

**Authors:** Mengjuan Chen, Xinyao Wang, Yang Liu, Pao Li, Rongrong Wang, Liwen Jiang

**Affiliations:** 1College of Food Science and Technology, Hunan Agricultural University, Changsha 410128, China; 2Hunan Provincial Key Laboratory of Food Science and Biotechnology, Hunan Agricultural University, Changsha 410128, China; 3Hunan Agricultural Product Processing Institute, Hunan Academy of Agricultural Sciences, Changsha 410125, China

**Keywords:** yellow lantern pepper sauce, discoloration, carotenoids, physiochemical indices, fermentation

## Abstract

Color is one of the important indicators affecting the quality of fermented pepper sauces, and it is closely related to carotenoid composition. This study systematically analyzed the changes in carotenoids and related physiochemical indices during the fermentation of yellow lantern pepper sauce. The CIELab color values indicated that *L** and *C** displayed a significant decreasing trend during fermentation. After 35 days of fermentation, the total carotenoid content significantly reduced from 3446.36 to 1556.50 μg/g DW (*p* < 0.05), and the degradation rate was 54.84%. Among them, the total content of carotene decreased by 56.03% during fermentation, whereas the degradation rate of xanthophylls and their esters was 44.47%. According to correlation analysis, violaxanthin myristate and lutein played a pivotal role in *L**, *a **, *b **, chroma (*C**), and yellowness index (*YI*). Moreover, PCA analysis indicated that lactic acid and acetic acid were the important qualities affecting the stability of pigment in fermented yellow lantern pepper sauce, which might also be the inducement of the color change. This work gives additional information concerning the discoloration of yellow lantern pepper sauce during fermentation and provides theory evidence regulating and improving the sensory qualities of yellow lantern pepper sauce.

## 1. Introduction

Yellow lantern pepper (*Capsicum chinense* Jacq.), a perennial herb of Solanaceae and *Capsicum* genus, is a unique pepper variety in Hainan province, China [1]. They are rich in nutrients and bioactive compounds, including vitamin C, carotenoids, capsaicinoids, and mineral compositions. Yellow lantern pepper has a spiciness of about 150,000~170,000 SHU, which is higher than most pepper species [2]. Thus, yellow lantern peppers are usually made into pepper sauce to reduce their spiciness. Yellow lantern pepper sauce has a gorgeous yellow color and strong fragrance, which is popular with the spicy crowd [3]. Usually, natural fermentation and inoculation fermentation are the main ways to prepare pepper sauce. However, compared to natural fermentation, inoculation fermentation can shorten the fermentation time, improve the fermentation efficiency, effectively avoid contamination by adverse bacteria, and ensure the stability and safety of the fermentation process. Hence, inoculated fermentation has drawn increasing attention to the pepper sauce process.

Generally, color is highly relevant in the total quality assessment of food products, and color analysis by sensory testing has been done for quality control of food production [4]. Color is an important quality index defining the sensory properties of pepper sauce. Consumers’ demands and competition imposed by the market have forced the provision of high-standard products with better sensory characteristics. For fresh peppers, carotenoids are the major pigments, the presentation and deterioration of color quality are mainly related to carotenoids [5]. Carotenoids are lipid soluble and mainly composed of C40 polyene chain skeletons with different colors (yellow, orange, and red) due to conjugated double bonds [6]. Generally, carotenoids in fruits are located in chromoplasts, and their composition is affected by the variety, cultivation environment, maturity, picking season, processing technology, and other qualities [7,8,9,10]. In previous studies, the processing is an important factor affecting the color and the composition of carotenoids in foods. Kaur and Kaur (2020) found that the total carotenoid content in the sweet pepper puree was negatively affected during the processing, and its color parameters changed significantly [11]. Stinco et al. (2019) showed that the color differences (ΔE*ab) of carotene juice treated by high-pressure processing and its carotenoid content both increased with the increase of the pressure or processing times [12].

Current studies mainly focused on the changes of carotenoids in plant-derived food during thermal processing, high-pressure processing, and drying [13,14,15]. Relatively little research is about the effect of fermentation on its stability. In a study conducted by Kun et al. (2018) they investigated the impact of fermentation on the carrot juice inoculated with three different *Bifidobacterium* strains [16]. Noteworthy was the degradation of α-carotene and β-carotene in the juice was between 15 and 45% depending on the strain. Oloo et al. (2014) evaluated the impact of lactic acid bacteria (LAB) fermentation on the levels of β-carotene in orange-fleshed sweet potatoes, and the retention of β-carotene was 93.97% after fermentation [17]. Indeed, carotenoids include precursors of vitamin A (VA) and those that cannot be converted to the VA (most notable lutein). All of them show strong antioxidant properties and involve in disease prevention and health promotion [18]. However, food processing can improve the bioaccessibility of carotenoids and increase the chances of absorption and utilization [19]. Therefore, the changes in carotenoids in plant-derived food during processing are attracting increased interest.

As a very popular traditional fermented food in China, the yellow lantern pepper sauce has a fascinating bright yellow color, but it frequently undergoes color deterioration during fermentation. Visual characteristics (color) determine anticipation of products before purchasing, further affecting the perception of other sensory attributes [20,21]. Therefore, the discoloration of fermented yellow lantern pepper sauce may be perceived as product deterioration by consumers and reduce purchasing desire. It is necessary to study the color deterioration of fermented pepper sauce and provide data support for the regulation and promotion of production quality. However, little literature reports the changes in carotenoids and their related qualities during the fermentation of peppers.

The aim of this study was to investigate the color changes in the yellow lantern pepper sauce during fermentation. The causes of discoloration were comprehensively analyzed from carotenoid composition profiles, and the correlation between color and carotenoid composition was explored. Besides, the changes in physiochemical indices during fermentation were also determined, and their effects on the stability of carotenoids were analyzed. The study may provide theory evidence regulating and improving the sensory qualities of yellow lantern pepper sauce.

## 2. Results and Discussion

### 2.1. Color Parameters

The changes in color during fermentation are presented in Table 1. The *L**, *a**, and *b** decreased during fermentation, especially *L** (from 45.43 to 33.82) and *b** (from 51.73 to 29.04), indicating that the color and luminosity of the pepper sauces deteriorated with the extension of fermentation time. The chroma (*C**) and yellowness index (*YI*) are mainly used to quantify the brighter and yellow degree of food [22]. As shown in Table 1, the *C** and *YI* values also decreased obviously during fermentation. Combined with the changes of *L**, it was found that the overall color of the pepper sauce went from bright yellow to brownish-yellow during fermentation. It was speculated that the enzymes (lipase, nitrite reductase etc.) and secondary metabolites (bacteriocins, organic acids, etc.) produced by LAB affected the rate of fermentation, plant tissue state, and water holding capacity, thus manipulating the color characteristics of the final product [23,24]. A similar phenomenon was also reported in the case of fermented cashew apple juice [25].

### 2.2. Carotenoid Profile

A total of 47 carotenoids were detected in all samples, including 14 free carotenoids and 33 esterified carotenoids (Appendix A). The total carotenoid content of the sample at day 0 reached 3446.36 μg/g DW, and it was higher than that in many yellow and orange peppers reported in previous literature [26,27], indicating that yellow lantern pepper could be used as a high-quality natural source of carotenoids in daily diet. The degradation rate of total carotenoid content increased (from 26.21% to 54.84 %) during fermentation, especially during 14–21 d (increased from 28.53 % to 35.52%). After 35 days of fermentation, the total carotenoid content was significantly reduced to 1556.50 μg/g DW (*p* < 0.05), and the degradation rate was 54.84 %. The results indicated that fermentation could affect the stability of carotenoids to some extent. 

The total content of carotene reached 3065.59 μg/g DW in the sample at day 0, accounting for 88.95% of the total carotenoid content. However, the total content of carotene decreased by 56.03% after fermentation. Among the carotenes, (*E/Z*)-phytoene was the main component, and its content (2940.00 μg/g DW) was higher than that reported in the existing literature [28,29]. It could be related to pepper varieties and maturity [30,31]. With the extent of fermentation time, the content of (*E/Z*)-phytoene decreased continuously (Figure 1A); it was 55.78% lower than that of the sample at day 0 when fermented for 35 days. As shown in Figure 1, (*E/Z*)-phytoene is an important intermediate substance in the carotenoid metabolism pathway and exhibits strong antioxidant properties and instability [32]. It can be dehydrogenated and transformed through multiple steps of the enzyme system in plants, such as phytoene dehydrogenase (PDS), δ-carotene dehydrogenase (ZDS), ε-lycopene cyclase (LCYE) and β-lycopene cyclase (LCYB) to form δ-carotene, β-carotene and α-carotene [33]. Hence, it could explain the increase of α-carotene and β-carotene content at fermentation metaphase (Figure 1B), and also indicated that carotenoid-related enzymes in peppers still showed activity during fermentation. Interestingly, the levels of β-carotene were higher than that of α-carotene during fermentation, which may be due to the differences in the inhibition of LCYE and LCYB activities.

Xanthophylls in the sample at day 0 are composed of the free and esterified forms, and the content was 87.52 μg/g DW and 296.25 μg/g DW, respectively. The main free xanthophyll in yellow pepper was lutein during the whole fermentation time, and it was significantly higher than other free xanthophylls. The content of lutein decreased gradually from 69.55 μg/g DW in the sample at day 0 to 33.90 μg/g DW after fermentation, and the retention rate was 48.74% (Appendix A). However, free xanthophylls in plant tissues are often combined with medium and long-chain fatty acids to form stable xanthophyll esters [34,35]. In all the samples, 10 violaxanthin esters, 9 zeaxanthin esters, 6 lutein esters, 3 β-cryptoxanthin esters, 3 rubixanthin esters, 1 neoxanthin ester, 1 antheraxanthin ester were detected (Appendix A). Thereinto, violaxanthin esters (124.02 μg/g DW), zeaxanthin esters (77.87 μg/g DW), and lutein esters (55.11 μg/g DW) were the major xanthophyll esters in yellow lantern pepper (Figure 2), which in accordance with previous researches on yellow peppers [36]. The level of total xanthophyll esters dropped by 43.08% after fermentation. After 35 days of fermentation, except for violaxanthin esters, the degradation rates of zeaxanthin esters and lutein esters were lower than that of zeaxanthin and lutein. Interestingly, the degradation rate of antheraxanthin ester was the highest (100%) among the xanthophyll esters, and it was not detected in yellow lantern pepper sauces. 

Generally, carboxylic acids can be produced during fermentation and provide an acidic environment for fermented pepper sauces. Xanthophyll esters are unstable under acidic conditions and can be hydrolyzed into free xanthophylls and fatty acids, and further transform or degrade. Interestingly, in Appendix A, not all the xanthophylls esters continually decreased during the whole fermentation, such as violaxanthin-myristate-caprate, violaxanthin-myristate-laurate, lutein dimyristate and violaxanthin dimyristate (Figure 2). Among them, violaxanthin dimyristate was only detected in all the samples at the fermentation stage. The increase in xanthophyll esters content may be due to the reversible hydrolysis of the esters under acidic conditions or the involvement of lipase secreted by microorganisms in the ester synthesis reaction during fermentation [37]. 

### 2.3. Physiochemical Properties

#### 2.3.1. Crude Fat and Free Fat Acid

Crude fat is a general term for all fat-soluble substances and contains a variety of nutritional and functional components, such as carotenoids. Hence, the changes in lipid composition could influence the stabilization of carotenoids [38]. In Table 2, the crude fat content decreased from 52.76 to 15.31 g/kg DW during fermentation (71.02% decrease). Actually, the microorganisms secrete lipases during fermentation, which can degrade the fat into FFAs [39]. As shown in Table 2, the FFAs content significantly increased to 32.06 g/kg DW (*p* < 0.05) at day 35. As flavor precursors, FFAs play a key role in the flavor formation of pepper sauce during fermentation [40]. Moreover, the increase in FFAs content is also related to the hydrolysis of xanthophyll esters, which could reduce their stability to some extent.

#### 2.3.2. pH Value and Organic Acids

The changes in pH value and organic acids during fermentation are shown in Table 2. The organic acids could impact the balance of flavor, and inhibit the growth of undesirable microorganisms by reducing pH during fermentation [41]. However, carotenoids are easily degraded during fruit and vegetable processing since they are sensitive to pH value [42]. In the sample at day 0, oxalic acid, malic acid, citric acid, tartaric acid, and succinic acid were detected, and the total organic acid content was 4.18 mg/g fresh weight (FW). Oxalic acid was the main organic acid in the sample at day 0 with 2.93 mg/g FW, and it is also widely found in other plant-based foods and has a negative impact on the taste of food. However, fermentation could affect the compositions of organic acids, lactic acid, and acetic acid found in fermented pepper sauces. For fermented pepper sauces, oxalic acid, malic acid, lactic acid, and acetic acid were the dominant organic acids after fermentation, and their content respectively reached 2.53, 1.62, 11.10, and 1.55 mg/g FW. The content of oxalic acid decreased significantly (*p* < 0.05) during fermentation, which was consistent with the previous results in fermented common purslane and mango slurries [43,44]. In addition, lactic acid increased at day 7 (2.82 mg/g FW), while acetic acid was detected until day 21 with 0.70 mg/g FW. This might be related to the pH during fermentation. The sugars in peppers could be used by *Lactobacillus spp.* to produce lactic acid, resulting in a decrease in pH (from 5.70 to 4.16). The acid environment was more conducive to the growth and metabolism of acetic acid bacteria and further increased the content of acetic acid. Meanwhile, the contents of citric acid, tartaric acid, and succinic acid remained at a low level with little fluctuation in the whole fermentation process. Overall, the total contents of seven organic acids reached 17.52 mg/g FW after fermentation, and the pH value decreased to 4.16. However, the taste of the yellow lantern pepper sauce on day 35 of fermentation was slightly sour, and the pH was lower than 4.5 (pH parameter for similar products) [45]. 

### 2.4. Correlation between Color Parameters and Carotenoids

Generally, the inartificial free carotenoids synthesized in plants are all-trans isomers [46]. However, some carotenoids are unresistant to pH and can suffer isomerization cis/trans of their double bonds. The trans form of carotenoids presents a darker color, and the cis bonds present a weakening of color [47]. In order to investigate the effect of carotenoids on color, correlation and cluster analysis were used to investigate the relationship between carotenoid content and color during fermentation (Figure 3). According to the results, 10 kinds of carotenoids were significantly correlated with *L**, *a **, *b **, *C**, and *YI* (*p* < 0.05), including violaxanthin myristate, lutein, lycopene, (*E/Z*)-phytoene, etc. Among them, violaxanthin myristate and lutein were extremely significant positive correlations with color (*p* < 0.01), and they were closely related to the color quality of fermented pepper sauces. In addition, according to the results of cluster analysis, *C** and *b** were in the same cluster, indicating that the change of *C** was mainly affected by *b** during, followed by *L**. In processed pumpkin and carrot products, it also was proved that there was a direct correlation between carotenoid content and sample color, and lutein played a crucial role in the color among carotenoid monomers [48,49]. 

### 2.5. PCA Analysis of Physiochemical Indices and Carotenoids

PCA analysis was conducted to screen out the important physiochemical indices affecting carotenoids content. In the bi-plot (Figure 4), the fermented pepper sauce samples presented a major difference from the sample at day 0. This phenomenon was due to the chopping process and fermentation environment. The degradation of carotenoids was related to the processing conditions, storage time, and temperature. With the progress of fermentation, fermented pepper sauce samples presented a regular tendency to move to the negative sides of Factors 1 and 2. It might be caused by the decrease in crude fat content, as well as the increase of FFAs and organic acids. Citric acid, oxalic acid, and pH were positively correlated with carotene and most xanthophylls and their esters but negatively correlated with lactic acid, acetic acid, succinic acid, and FFAs. Besides, crude fat, lutein, and its esters were close to each other in Figure 4, representing a strong correlation between them. The overlapping information is shown in Appendix A. Similarly, from the correlation coefficient (r) and *p* value, lactic acid and acetic acid present an extremely significant negative correlation with lycopene (r = −1.000 and −0.941), lutein (r = −0.943 and −0.941), (*E/Z*)-phytoene (r = −1.000 and −0.941), etc. These results implied that the increase of lactic acid, acetic acid, succinic acid, and FFAs were closely related to the stability of carotenoids, and might further affect the color of fermented yellow lantern pepper sauce during fermentation. Hence, regulating the fermentation process is of great significance to stabilize the color and quality of fermented pepper sauce respectively.

## 3. Materials and Methods

### 3.1. Bacterial Cultures

*Lactobacillus plantarum* was purchased from the China Center of Industrial Culture Collection (CICC), and the deposit number of *Lactobacillus plantarum* is CICC No. 20265. The preliminary tests proved that this strain had strong salt tolerance (10% NaCl) and acid production ability (reducing the pH of the culture medium from 6.38 to 3.13 within 24 h), which satisfied the fermentation demand. They were inoculated into MRS broth medium (Guangdong Huankai Microbial Sci & Tech. Co., Ltd., Guangdong, China), grown, and expand at 37 °C for 24 h. The microbial cells were obtained after centrifugation at 3000 g for 10 min, then the fermented seed liquor was prepared with a cell count of 10^8^ CFU/mL.

### 3.2. Preparation of Yellow Lantern Pepper Sauce

Yellow lantern pepper (*Capsicum chinense* Jacq.) was harvested in Wenchang city, Hainan province, China (longitude 108°21 ‘to 111°03’ east, latitude 19°20 ‘to 20°10 north). The fresh yellow lantern peppers were cleaned and dried, then chopped into small pieces (5 mm × 5 mm). Thereafter, 2% (*w/w*) of *Lactobacillus plantarum* at 10^8^ CFU/mL and 80 g/kg salt were added to the minced pepper and then stirred evenly. The samples were divided into 18 groups, about 300 g per group. They were put into pickle jars and sealed with water to exclude air and then fermented at room temperature. The yellow lantern pepper sauce samples were obtained at 0, 7, 14, 21, 28, and 35 days, respectively. The collected sauces were immediately taken to measure the pH value, organic acids, and color. Another 150 g sauces were quickly frozen with liquid nitrogen, then freeze-dried, ground into powder, and stored at −80 °C for further use; this process was under dark conditions. The preparation process of yellow lantern pepper sauce is shown in Figure 5.

### 3.3. Color Analysis

The color of the pepper sauce samples was measured with a precise color reader (WR-18, Shenzhen Wave Optoelectronics Technology Co., Ltd., Shenzhen, China) using a D65 light source and at an observed angle of 10°. The pepper sauce sample was mixed with an equal volume of 1 g/kg vitamin C solution for antioxidant and homogenized [50], added into a colorimetric dish with a light diameter of 10 mm, then measured under reflection mode. The CIELAB *L**, *a**, and *b** values were obtained at 5 different points and averaged as measurement data, and 3 replicates for each sample. The chroma (*C**) and yellowness index (*YI*) were calculated using the following equations:C*= (*a**^2^ + *b**^2^) ^0.5^(1)
YI = 142.86 *b**/*L**(2)

### 3.4. LC-MS/MS Analysis of Carotenoids

Freeze-dried samples (50 mg) were extracted with 0.5 mL of n-hexane/acetone/ethanol mixture containing 1 g/L BHT (1:1:1, by volume). After 20 min of the vortex at room temperature, they were centrifuged at 7200 g for 5 min at 4 °C, the supernatant was taken out and the above steps were repeated three times. All supernatant was collected and dried under nitrogen flux, then reconstituted using 100 μL methanol/methyl tert-butyl ether mixed solution (1:1, by volume). After filtering by nylon membrane (0.22 μm), the reconstituted solution was stored in a brown injection bottle for LC-MS / MS analysis.

Carotenoids were analyzed with the Ultra Performance Liquid Chromatography (ExionLC™ AD, Shanghai AB SCIEX Analytical Instrument Trading Co., Shanghai, China) and a Tandem Mass Spectrometer (QTRAP^®^ 6500+, Shanghai AB SCIEX Analytical Instrument Trading Co., Shanghai, China). Separation was carried out on a C30 YMC column (3 μm, 100 mm × 2.0 mm i.d.) (YMC Co., Ltd., Tokyo, Japan). The mobile phases comprised of methanol /acetonitrile = 1:3 (0.01% BHT and 0.1% formic acid) (eluent A) and methyl tert-butyl ether (0.01% BHT) (eluent B) at an injection volume of 2 μL. The flow rate was 0.8 mL/min. The linear gradient program was as follows: from 0 to 3 min 0% B; from 3 to 5 min, 0 to 70% B; from 5 to 9 min, 70 to 95% B; from 9 to 10 min,95 to 0% B; from 10 to 11 min 0% B. The Atmospheric Pressure Chemical Ionization Source temperature was 350 °C and Curtain Gas was 25 psi. In Q-Trap 6500+, each ion pair is scanned according to the optimized Delustering Potential and Collision Energy [51,52]. The free carotenoid standard was configured with a methyl tert-butyl ether/methanol mixed solution as 1 mg/mL stock solution and then diluted to an appropriate concentration for subsequent use. The esters were quantified using the curves from their corresponding carotenoids, and the carotenoid standards regression equations are listed in Appendix A.

### 3.5. Determination of pH Value and Organic Acid

The pH value was determined by a pH meter (PHS-25; INESA Scientific Instrument Co., Shanghai, China). 

The organic acid content was measured by Ye et al. (2018) with slight modification [53]. 2.5 g pepper sauces sample was placed in a 50 mL centrifuge tube and then added 30 mL ultrapure water. After vortex oscillation and centrifugation, the supernatant was transferred to a 100 mL volumetric flask, and the operation was repeated twice. Then, the solution was injected into a 2 mL liquid phase vial after filtration through a 0.22 μm PES membrane. Organic acids were analyzed with an HPLC system (e2695, Waters Corporation, Milford, MA, USA) equipped with an oven for controlling the column temperature (30 °C), a UV-Vis detector (2996, Waters Corporation, Milford, MA, USA), and an Agilent Polaris C18-A chromatographic column (4.6 mm × 250 mm, 5 μm). The mobile phase consisted of a potassium dihydrogen phosphate solution of 0.01 M (pH 2.2) and methanol (97:3, by volume). Isocratic elution was applied to the mobile phase with a flow rate of 0.5 mL/min. A wavelength of 210 nm was selected for quantification. The organic acid standard regression equations are listed in Appendix A.

### 3.6. Determination of Crude Fat and Free Fatty Acids (FFAs)

The crude fat of freeze-dried samples was determined by the Soxhlet method, and the measurement was described by Mohamed Ahmed with slight modification [54]. The extraction was carried out in a Soxhlet apparatus at 50 °C for 6 h using petroleum ether as the extraction solvent. After the extraction, the solvent was evaporated under reduced pressure and the content of crude fat was calculated.

The crude fat extract was fully dissolved with 40 mL of solution (ether: 95% ethanol = 1:1, by volume), then a few drops of phenolphthalein indicator were added. Sodium hydroxide solution (0.1 mol /L) was used to titrate until the solution turns a reddish color. The FFA content was calculated based on the consumption of sodium hydroxide solution (based on oleic acid) [55]. The calculation formula was as follows: X(g/kg) = (0.1 × v × 282)/m(3)

In the formula, v represented standard NaOH volume (mL), m represented the quantity of crude fat (g), and X represented FFA content.

### 3.7. Statistical Analysis

All the measurements were carried out in triplicate independently, and three samples were analyzed for each treatment. The results were expressed as mean value ± standard deviation. Data were analyzed by one-way analysis of variance with Duncan’s multiple range test methods in SPSS 22.0 (SPSS Inc., Chicago, IL, USA), and *p* < 0.05 was classified as statistically significant. Clustering correlation heatmap with signs was performed using the OmicStudio tools at https://www.omicstudio.cn (accessed on 5 March 2022). SPSSAU (Version 20.0), an Online Application Software (https//www.spssau.com, accessed on 8 March 2022), was used for visual analysis of the Correlation Heatmap based on Spearman. Origin 2021b (OriginLab Corporation, Northampton, MA, USA) was used for plotting. Principal components analysis (PCA) was used to further analyze the carotenoid content and color data by Unscrambler X software (version 10.4, Camo Inc., Norway).

## 4. Conclusions

Color features are one of the most important quality traits of fermented pepper sauces, which not only affect the perception of other sensory attributes but also determine the purchasing desire. Our study, aimed at investigating the color and carotenoid composition changes in the yellow lantern pepper sauce during fermentation, and screened out physicochemical indices remarkably related to carotenoid stability. During fermentation, the *L**, *a **, *b **, *C**, and *YI* values all decreased, indicating that the yellowness, brightness, and brilliance of yellow lantern pepper sauce were significantly affected. The total carotenoid content of the sample at day 0 reached 3446.36 μg/g DW, and it significantly reduced to 1556.50 μg/g DW after fermentation (*p* < 0.05). The degradation rate of total carotenoid content increased (from 26.21% to 54.84 %) during fermentation, especially during 14–21 d (from 28.53 % to 35.52%). Compared with carotene, the xanthophylls and their esters showed higher stability throughout the fermentation process. Due to the decrease in carotenoid content, the color characteristics and sensory quality of yellow lantern pepper sauce all deteriorated. The PCA analysis showed that violaxanthin myristate and lutein played the most important effect on color, and lactic acid, acetic acid, succinic acid, and FFAs were the important qualities associated with the stability of pigments in fermented yellow lantern pepper sauces. From the aspects of carotenoids degradation rate and related physiochemical indices, it is appropriate to ferment pepper sauce for about 14 days. This research would provide a theoretical basis for improving the sensory qualities of yellow lantern pepper sauce.

## Figures and Tables

**Figure 1 molecules-27-07139-f001:**
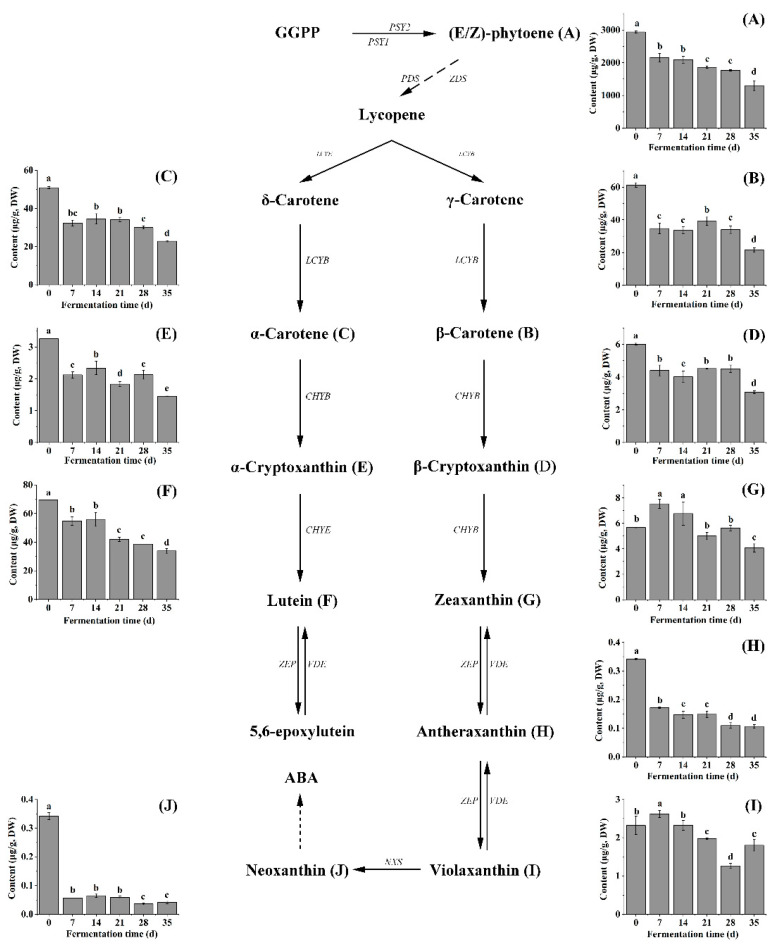
Changes in the content of carotenoids during fermentation. (**A**) (E/Z)-Phytoene. (**B**) β-Carotene. (**C**) α-Carotene. (**D**) β-Cryptoxanthin. (**E**) α-Cryptoxanthin. (**F**) Lutein. (**G**) Zeaxanthin. (**H**) Antheraxanthin. (**I**) Violaxanthin. (**J**) Neoxanthin. The different letters in the same figure indicate significantly different (*p* < 0.05).

**Figure 2 molecules-27-07139-f002:**
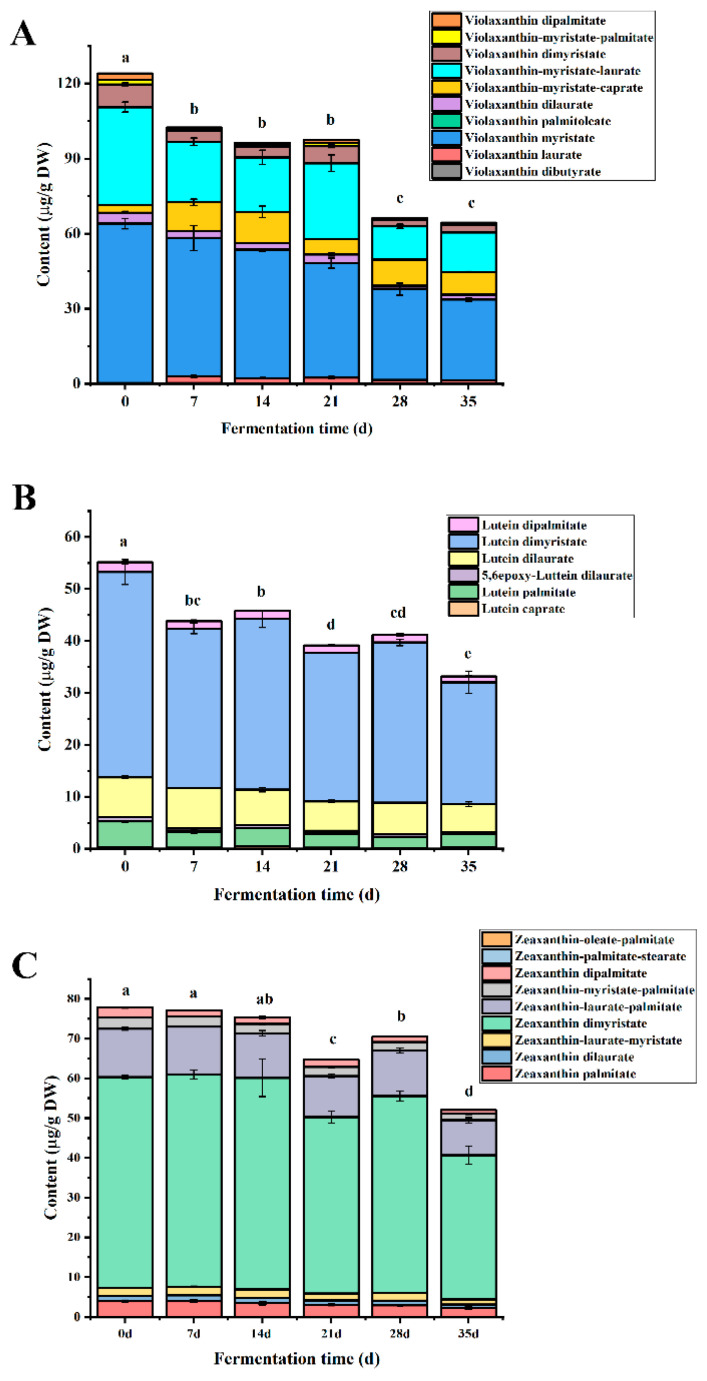
Changes in the content of (**A**) violaxanthin esters, (**B**) lutein esters and (**C**) zeaxanthin esters during fermentation. The different letters in the same figure indicate significantly different (*p* < 0.05).

**Figure 3 molecules-27-07139-f003:**
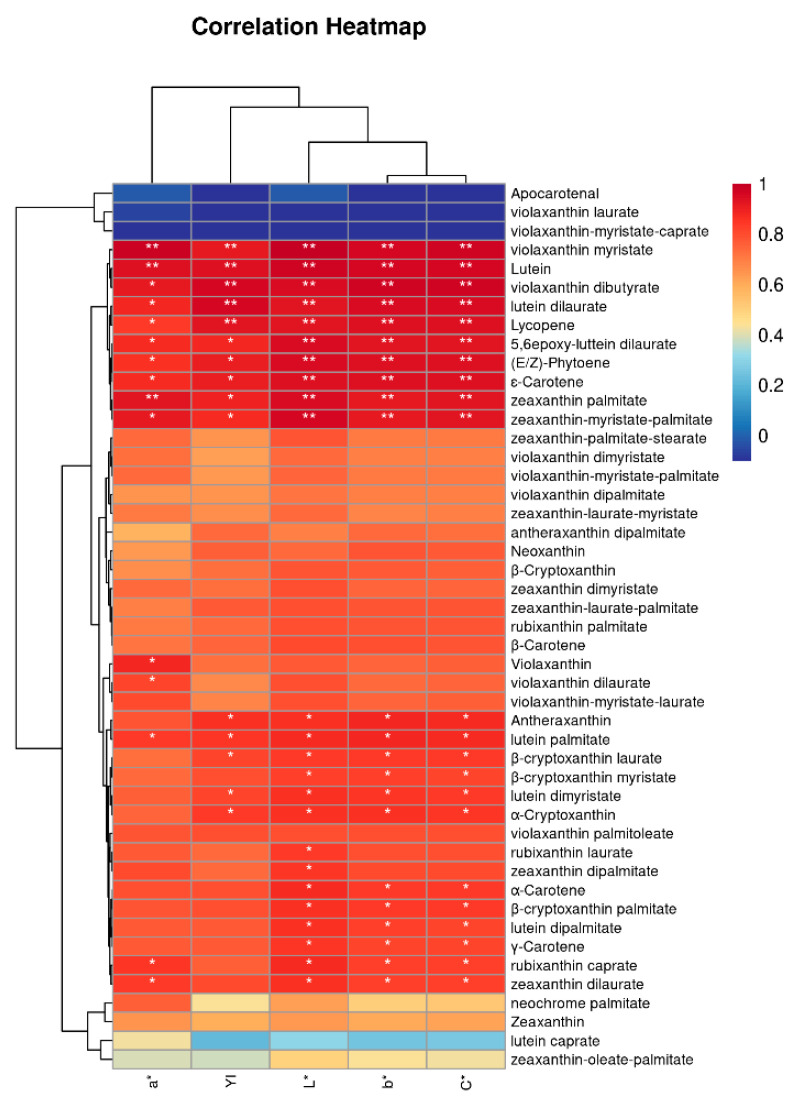
Correlation of color attributes and carotenoids content. “*” means significant correlation (*p* < 0.05); “**” indicates extremely significant correlation (*p* < 0.01).

**Figure 4 molecules-27-07139-f004:**
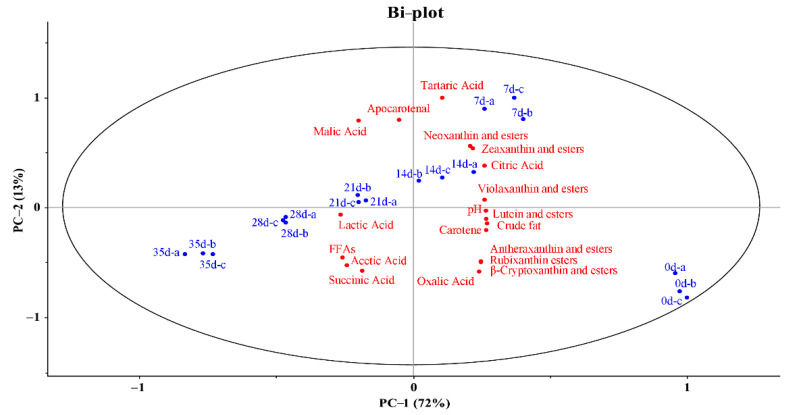
PCA analysis of fermented yellow lantern pepper sauce samples. The red letters represented physiochemical indices; the blue letters represented different carotenoids.

**Figure 5 molecules-27-07139-f005:**
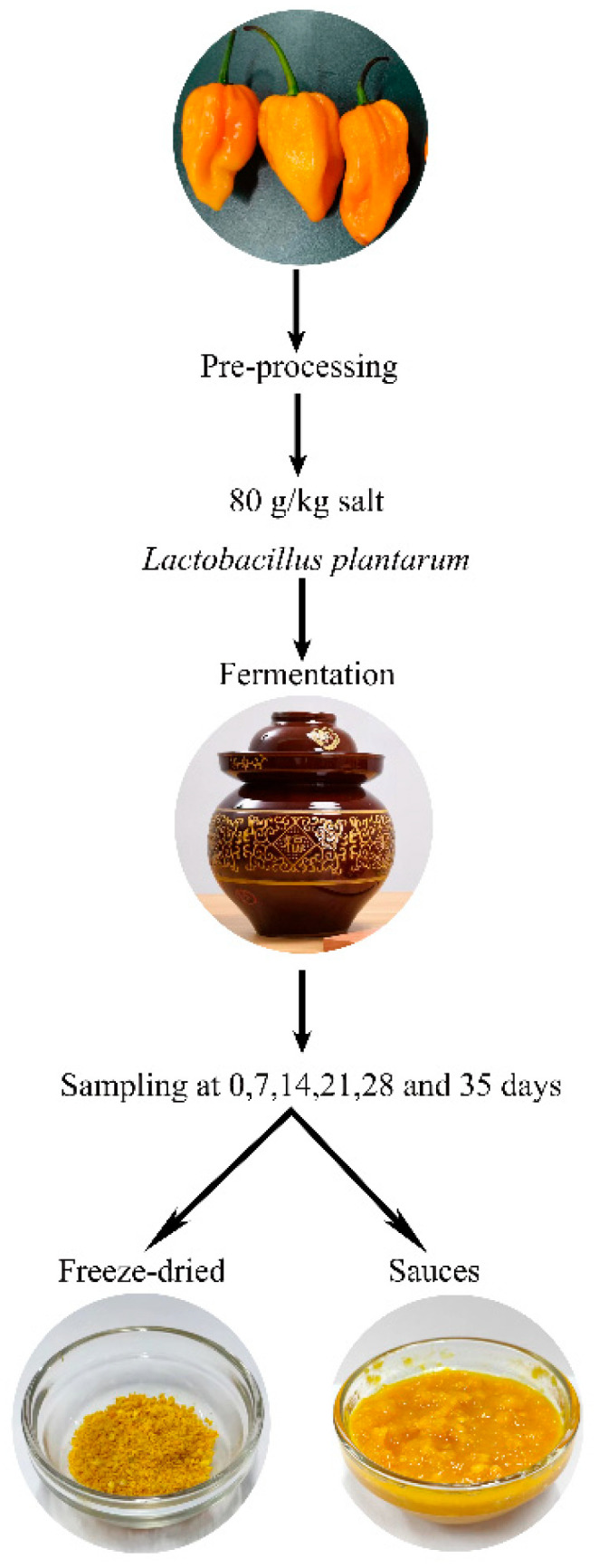
The preparation process of yellow lantern pepper sauce.

**Table 1 molecules-27-07139-t001:** Changes in the color parameters during fermentation.

	0 d	7 d	14 d	21 d	28 d	35 d
*L**	45.43 ± 1.72 ^a^	42.51 ± 1.09 ^b^	40.45 ± 0.88 ^c^	37.89 ± 1.09 ^d^	35.72 ± 0.86 ^e^	33.82 ± 1.68 ^f^
*a**	15.54 ± 0.73 ^a^	14.75 ± 0.49 ^b^	14.20 ± 0.58 ^bc^	13.98 ± 0.66 ^c^	13.67 ± 0.86 ^c^	13.48 ± 0.88 ^c^
*b**	51.73 ± 5.79 ^a^	45.51 ± 1.61 ^b^	39.03 ± 0.95 ^c^	33.07 ± 1.26 ^d^	31.54 ± 1.89 ^d^	29.04 ± 2.25 ^e^
*C**	53.82 ± 5.25 ^a^	47.94 ± 1.53 ^b^	41.52 ± 0.86 ^c^	35.87 ± 1.27 ^d^	34.36 ± 1.65 ^d^	32.07 ± 1.25 ^e^
*YI*	162.95 ± 20.09 ^a^	152.96 ± 4.31 ^b^	137.85 ± 2.41 ^c^	124.69 ± 3.83 ^d^	126.31 ± 9.25 ^d^	122.60 ± 5.75 ^d^

Values are expressed as means ± SD of 3 replicate; the different letters in the same row indicate significantly different (*p* < 0.05).

**Table 2 molecules-27-07139-t002:** Changes in the related physiochemical properties during fermentation.

	0 d	7 d	14 d	21 d	28 d	35 d
pH	5.70 ± 0.01 ^a^	5.19 ± 0.01 ^b^	4.70 ± 0.02 ^c^	4.32 ± 0.02 ^d^	4.19 ± 0.01 ^e^	4.16 ± 0.01 ^f^
Oxalic acid (mg/g FW)	2.93 ± 0.02 ^a^	2.64 ± 0.05 ^b^	2.56 ± 0.02 ^c^	2.51 ± 0.01 ^d^	2.50 ± 0.01 ^d^	2.53 ± 0.00 ^cd^
Tartaric acid (mg/g FW)	0.19 ± 0.03 ^bc^	0.36 ± 0.02 ^a^	0.22 ± 0.01 ^b^	0.13 ± 0.01 ^d^	0.13 ± 0.01 ^d^	0.18 ± 0.02 ^c^
Malic acid (mg/g FW)	0.30 ±0.03 ^d^	1.46 ± 0.07 ^c^	1.97 ± 0.03 ^a^	1.62 ± 0.03 ^b^	1.64 ± 0.03 ^b^	1.62 ± 0.03 ^b^
Lactic acid (mg/g FW)	N.D.	2.82 ± 0.02 ^e^	7.19 ± 0.13 ^d^	9.91 ± 0.37 ^c^	10.77 ± 0.01 ^b^	11.10 ± 0.05 ^a^
Acetic acid (mg/g FW)	N.D.	N.D.	N.D.	0.70 ± 0.04 ^c^	1.46 ± 0.03 ^b^	1.55 ± 0.03 ^a^
Citric acid (mg/g FW)	0.52 ± 0.01 ^a^	0.50 ± 0.01 ^a^	0.40 ± 0.03 ^b^	0.29 ± 0.01 ^c^	0.23 ± 0.03 ^d^	0.18 ± 0.02 ^e^
Succinic acid (mg/g FW)	0.24 ± 0.00 ^cd^	0.22± 0.03 ^d^	0.29 ± 0.01 ^b^	0.25 ± 0.01 ^c^	0.26 ± 0.00 ^c^	0.36 ± 0.00 ^a^
Crude fat (g/kg DW)	52.76 ± 0.37 ^a^	36.49 ± 0.14 ^b^	29.28 ± 0.64 ^c^	24.23 ± 0.55 ^d^	17.19 ± 0.76 ^e^	15.31 ± 0.36 ^f^
Free fat acid (g/kg DW)	22.20 ± 0.11 ^e^	22.48 ± 0.061 ^e^	24.20 ± 0.01 ^d^	27.31 ± 0.29 ^c^	29.36 ± 0.23 ^b^	32.06 ± 0.17 ^a^

Values are expressed as means ± SD of 3 replicate; the different letters in the same row indicate significantly different (*p* < 0.05).

## Data Availability

Data is contained within the article or Appendix A. The data presented in this study are available on request to the corresponding author.

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
