# Peer review of "Discoloration Investigations of Yellow Lantern Pepper Sauce (Capsicum chinense Jacq.) Fermented by Lactobacillus plantarum: Effect of Carotenoids and Physiochemical Indices"

_molecules, 2022, doi:10.3390/molecules27207139_

Round 1

Reviewer 1 Report

This article deals with the color change of yellow lantern pepper sauce, and its relationship with crucial compounds during storage. The findings are important for evaluating the useability of quality control of sauce production. But there are several issues which needs to be addressed before being considered for publishing.

1.     The description of fig.1 should be improved, it is difficult to understand the meaning of data change.  

2.     The relationship of sauce quality and color change (or carotenoids conc.) should be explained. Although the author already done the relationship of color change and carotenoids conc.

3.     In the conclusion, it seems no directly influence between carotenoids change and sensory qualities. The important of sensory qualities should be further discussed.

Reviewer 2 Report

The important ingredients of yellow lantern pepper sauce were analyzed completely during fermentation. It is of reference value. Color is defined in this manuscript as one of the important quality indicators of fermented products. However, there is no discussion on what standard of color should be maintained during fermentation, or the influence of color on sensory quality, etc. It is recommended to add the relevant discussion to the manuscript.

#1. There is no reference about the sensory quality (like color) for the yellow lantern pepper sauce in the introduction. Please add relevant references.

#2. The Heatmap analysis is based on the Spearman correlation. But the cluster analysis is also used when Heatmap analysis.

#3. Please add the discussion about the clusters analysis of carotenoids in the sauce in Figure 3.

#4. In P7 Line 218-221, is showed that the “violaxanthin myristate and lutein were extremely significant positive correlations with color (P < 0.01), and they could directly affect the color quality of fermented pepper sauces.”. Correlation analysis can only show whether there is a correlation between different components, and cannot be directly used to explain which component affects the other component. Please use more objective sentences to describe this result after referring to the contents of Ref. No. 47-48

#5. There are only scores plot and correlation loading plot showed in the Figure 4. The result of Bi-Plot in PCA can observe the correlation between samples and different components. Please add Bi-Plot and discuss further about the results presented in Bi-Plot.

#6. In P8 Line 244-247, it showed that the “These results implied that the increase of lactic acid and acetic acid were detrimental to the stability of carotenoids”. In fact, the main reason for the correlation between these two components should be the decrease of carotenoids and the increase of acids during the fermentation process. It is not recommended to make this inference directly, and please refer to #4 for modifying these sentences.

#7. In 2.3.2, the fermentation of Lactobacillus plantarum is the Homolactic fermentation. Other acids should not be produced during the fermentation process. Therefore, it is possible that the growth of acetic acid bacteria mentioned in the manuscript is due to the presence of other microorganisms in the raw material. If you want to avoid the production of acetic acid to affect the color of the product, is it necessary to sterilize the raw material? Please discuss more about this section.

#8. In P9 Line 279-280, it showed that “The pepper sauce sample was mixed with equal volume 1 g/kg vitamin C solution”. Why should vitamin C be added before analysis? Will the added vitamin C affect the color of the product? Please add the related reference and explain it.

#9. P11 Line 344, Dun-can’s multiple range test method should be Duncan’s multiple range test method.
